# The Use of Cultural Landscape Fragmentation for Rural Tourism Development in the Zemplín Geopark, Slovakia

Jana Rybárová [1], Radim Rybár [2], Dana Tometzová [1,*] and Gabriel Wittenberger [3]

1   Department of Geo and Mining Tourism, Institute of Earth Resources, Faculty of Mining, Ecology, Process
    Control and Geotechnologies, Technical University of Košice, 04200 Košice, Slovakia; jana.rybarova@tuke.sk
2   Department of Renewable Energy Sources, Institute of Earth Resources, Faculty of Mining, Ecology, Process
    Control and Geotechnologies, Technical University of Košice, 04200 Košice, Slovakia; radim.rybar@tuke.sk
3   Department of Montaneous Sciences, Institute of Earth Resources, Faculty of Mining, Ecology, Process Control
    and Geotechnologies, Technical University of Košice, 04200 Košice, Slovakia; gabriel.wittenberger@tuke.sk
*   Correspondence: dana.tometzova@tuke.sk

**Abstract:** This study outlines the creation of hiking routes in Slovakia's cultural landscape, focusing on regions with marginal interest, low tourism engagement, and predominant monocultural blocks. The methodology was systematically applied to the Zemplín Geopark in eastern Slovakia, drawing upon historical cartographic records from the Josephine mapping period (1764–1787) to the present day. The investigation identified and delineated 14 hiking trails, offering historical and tourism significance while promoting multifunctionality. Our research introduces sustainable development avenues for regions with marginal interest, providing ecological and tourist benefits that enhance the overall quality of life. The findings align with the Common Agricultural Policy's objectives for 2021–2027, addressing challenges related to large-scale field fragmentation. Two identified obstacles include property-legal challenges and issues arising from inadequate map registration, which current methods, unfortunately, fail to address.

**Keywords:** rural tourism; sustainable tourism; cultural landscape; hiking trails; land fragmentation; land use; Zemplín Geopark

## 1. Introduction

Rural tourism is an activity that represents an opportunity for economic stabilization and enhancement of sustainability for peripheral (rural) regions [1,2]. In 2020, this intention was endorsed by the declaration of the UNWTO (World Tourism Organization) designating it as the Year of Tourism and Rural Development. The WTTC and UNWTO proclaimed that destinations most severely affected require immediate and sustainable recovery [3].

The rural landscape, when considered as an object of tourism, should also be examined from an agricultural perspective. Generally, rural landscapes are perceived as cultural (anthropogenic) landscapes, shaped by the interplay of natural and human forces over time and space [4]. Several authors in the past have delved into defining the complexity of cultural landscapes, including Schmithüsen [5], Uhlig [6], Schwind [7], Ružička [8], Jäger [9], Wöbse [10], and Harteisen [11], among others.

Currently, tourism is emerging as a primary objective in numerous political strategies aimed at mitigating the risks associated with disasters and enhancing financial and economic stability. Slovakia, as a member of the United Nations, is fulfilling the commitments set out in the 17 goals of Agenda 2030 above the EU average, thereby aiding in the attainment of sustainable development objectives within the EU.

Slovakia is perceived as an agricultural country, with up to 60% of its population residing in countryside areas. On one hand, Slovakia has been shaped by the evolution of human society, including the modernization of production methods, specialization, concentration of agricultural production, and intensification. On the other hand, societal changes,

influenced by globalization and the decline of local economies, have impacted Slovakia, along with other European countries, especially after World War II. Further changes were a consequence of centrally planned economic management, such as the collectivization within Czechoslovakia, occurring in two stages (the first stage from 1949 to 1955 and the second stage from 1955 to 1960). This process involved consolidating land into unified agricultural cooperatives (in Slovakia known as JRD—Jednotné roľnícke družstvá) [12]. Despite the transformation of agricultural cooperatives after the fall of the totalitarian regime in 1989 [13], the extensive nature of fields has essentially remained unchanged.

The paradox of contemporary land management lies in the extreme fragmentation of agricultural land ownership, prevalent both in Slovakia and other countries in Europe, manifesting itself as a notable deficiency [14–16]. This phenomenon constitutes a negative and constraining factor for rural tourism development [17]. Its roots can be traced back to historical inheritance practices, as identified in the work of Sklenička et al. [18]. T. van Dijk [17], Sklenička [19], and Vranken & Swinnen [20] have pinpointed the issue of uneven land distribution among heirs over several centuries, resulting in a complex pattern of functionally inefficient parcels.

These remnants of historical land management, contributing to the configured landscape, represent an average field size of 12 ha in Slovakia, contrasting with the EU average of 3.9 ha (comparing with bordering countries: Hungary: 8.5 ha; Poland: 3.7 ha; Austria: 2.8 ha; Czech Republic: 9.5 ha) [21,22]. Examples of border regions in Slovakia and Austria illustrating the differences in landscape structure are depicted in Figure 1.

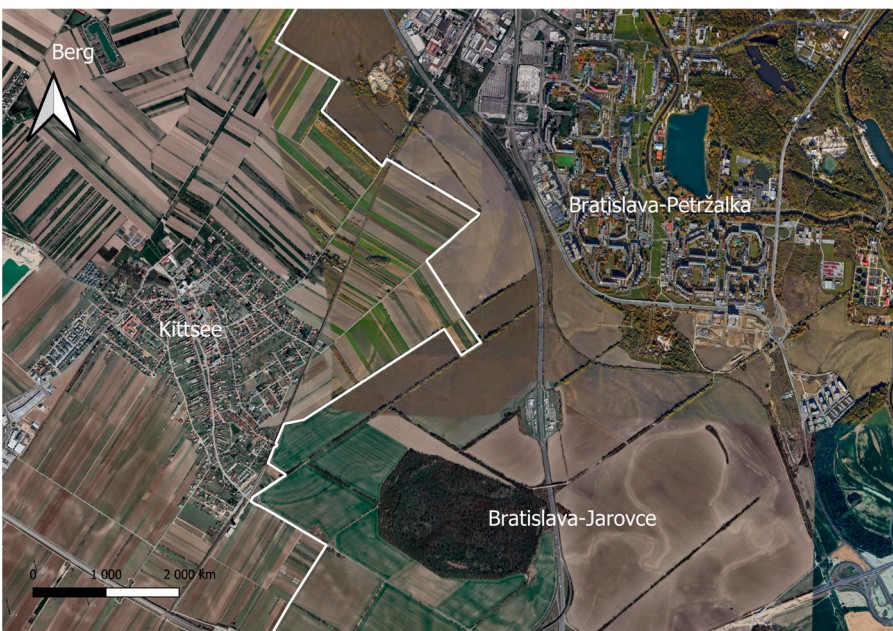

**Figure 1.** Illustrations of border territories. Austria (AT) is depicted on the left, while the surroundings of Bratislava, Slovak Republic, are shown on the right. The border between the countries is marked in white. Source: Ministry of Environment of the Slovak Republic website [23].

For this reason, the Slovak Republic faces challenges in implementing further strategic measures outlined in the Common Agricultural Policy (CAP) for the years 2021–2027, which the country intends to fulfill through the standard DZES (Good Agricultural and Environmental Land Condition) 7d standard [24]. The challenge for the Slovak Republic is its inclusion among the nations exhibiting the least landscape fragmentation within the EU [25].

Based on the specific quantitative indicators that are discussed below (2. Literature Review section), the Slovak Environmental Agency (SAŽP—Slovenská agentúra životného prostredia) [26] and European Environment Agency (EEA) [27] provide assessments of the landscape fragmentation status in individual European countries. Countries in North-

western Europe (Luxembourg, Belgium Netherlands, France, and Germany) are the most fragmented (with mesh densities ranging from 42 to 135 per 1000 km$^2$). Mediterranean countries, such as Italy, Greece, and Spain, generally exhibit a moderate level of landscape fragmentation, although urban coastal areas often experience higher degrees of fragmentation. Slovakia, Romania, and the Scandinavian countries have fewer than 10 meshes per 1000 km$^2$. Countries with large, protected areas, areas of hills and mountains, will inevitably be less fragmented. An exception to the overall trend appears to be the United Kingdom, which has managed to maintain a relatively low number of meshes despite its high road and population density. However, most countries exhibit significant fluctuations in landscape fragmentation depending on their geography, the localization of urban areas, and transportation connections. Moreover, different countries have different cultural approaches to settlement, which can also influence the fragmentation level. There is a high correlation between population density, traffic road density, and landscape fragmentation [27,28].

From one perspective, the process of extreme fragmentation disrupts the structural connections between habitats, leading to reduced resilience and a diminished ability to provide ecosystem services and support biodiversity. The ecological indicators and their impact on biodiversity were discussed by Fahrig [29] and Llausàs & Nogué [30].

From another perspective, even a very low level of fragmentation could worsen the resilience of the landscape, making it susceptible to water and wind erosion, leading to soil depletion of organic matter. According to Ambos et al. [25], this contributes to an overall loss of biodiversity in the country. Within Slovakia, the addressed issue is legislatively regulated through the Green Infrastructure Strategy, part of the Territorial System of Ecological Stability (ÚSES) project, as per the Slovak Environmental Agency (SAŽP). The legislative framework and the position of the ÚSES are defined by several laws, as referenced in [26,31]. This issue is intended to be addressed in Slovakia as part of its preparations for the Strategic Plan of the CAP 2021–2027 [32].

Our primary intention is to provide an answer to the following question: how could the level of tourism in the selected region be increased considering political, ecological, and social factors? We endeavor to propose a solution to the issue of natural resource renewal and the establishment of sustainability in the studied area. We consider this as a means of expediting global environmental solutions that contribute to the strategic priorities of the Union, accelerate the economic growth of the Union, and support an innovative ecosystem. This aligns with the fulfillment of the UN Sustainable Development Goals and the objective to achieve climate neutrality in the Union by no later than 2050, in accordance with the Paris Agreement [33]. We seek to present a methodology addressing the farming sector as a component of the country's economic framework, involving the creation of hiking trails with added value in terms of cultural and historical significance. We provide a clearer explanation of our theoretical framework in Figure 2.

Achieving sustainable rural development necessitates an appropriate level of landscape fragmentation. Landscape fragmentation occurs when continuous ecosystems are divided into smaller units, typically due to the expansion of urban areas or transportation networks [28]. Based on this definition, it is essential to focus on quantitative metrics to assess the extent of landscape fragmentation, whether it is low or high. Jaeger et al. [34] mentioned numerous quantitative indicators (e.g., level of landscape division, effective mesh size–meff, or splitting index) for determining patterns in landscape fragmentation. The most frequently discussed quantitative indicator, mentioned in several scholarly works (Jaeger [35], Girvetz et al. [36], Jaeger et al. [37], and Penn-Bressel [38]), is the effective mesh size.

On the one hand, effective mesh size serves as an indicator of landscape connectivity, representing the extent to which movement across different regions is feasible. A larger meff value indicates greater landscape connectivity. On the other hand, the effective mesh density (seff) quantifies landscape fragmentation, reflecting the extent to which movement across various areas of the landscape is impeded by fragmentation geometry. It quantifies the efficient number of patches per 1000 km$^2$, essentially measuring the density of these patches. The value of seff is derived by dividing 1000 km$^2$ by meff, thus indicating the

quantity of meshes per 1000 km$^2$. Higher effective mesh density values indicate increased fragmentation caused by barriers within the landscape [39].

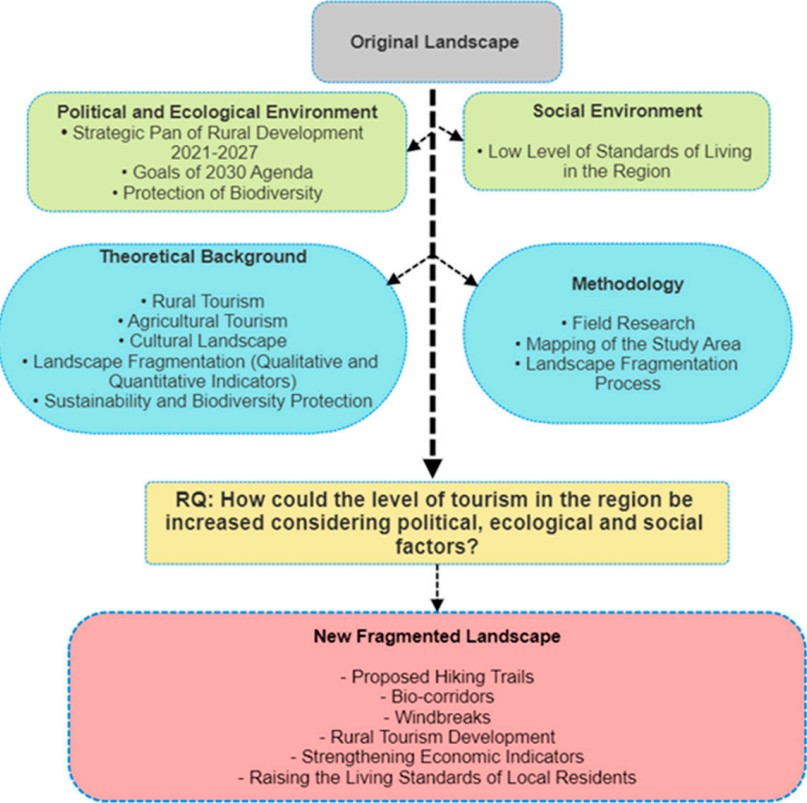

**Figure 2.** Framework diagram of the theoretical background.

As stated above, several authors have discussed the impact of fragmentation on biodiversity. However, within the realm of scientific knowledge, there is a gap concerning the impact of landscape fragmentation on rural tourism. Marginally, though, this topic has been addressed by Oueslati et al. [40] and Iannucci et al. [41].

This manuscript aims to further address this gap by examining the specific effect of landscape fragmentation on rural tourism dynamics, thereby contributing to a more comprehensive understanding of its implications for the tourism industry.

To date, there is no universally accepted delineation of rural tourism. The format of defining rural tourism is contingent on the level of examination. Rural tourism represents an alternative form of tourism, described by Lane & Kastenholz [42] as a means for the revival and preservation of rural zones. Lane [43] outlined five criteria for defining rural tourism: rural location, utility, scope, character, and spatial arrangement. Several systematic reviews have been conducted on defining rural tourism [44–46], synthesizing new insights from various existing studies. For instance, a study by Rosalina et al. [47] delves into a systematic quantitative literature review examining definitions of rural tourism and challenges within both developed and developing countries. Ekrem Aydin [48], through bibliometric mapping of data from the Web of Science Core Collection database for the period 2000–2020, revealed a trajectory of progress and potential areas for improvement. He identified high potential in the ecological development of rural areas. Additionally, academic researchers analyzed the connection between rural tourism and social, fiscal, and ecological aspects. According to them, seeking a comprehensive definition will result from the synergy of tourism and these three influences. The connection between tourism and its impact on economic levels was defined by Bănică & Camară [49]; Bulai et al. [50]; Ibanescu et al. [51]; Pascariu et al. [52], describing their findings with linear dependence. Other authors also concur with this approach [53–57]. Their intention was to define the

role of facilitating the connection between the revival of tourism and the revitalization of the economy [58]. Since 2004, it has been identified as one of the trends in rural tourism alongside its impact on the economic advancement of rural regions [59].

On the other hand, some argue that soft tourism [60,61] is a starting point for the long-term sustainable preservation of cultural heritage [62], especially for local communities. This could be a path to preserving local identity and the identity of the entire nation, on which a brand could be built, as suggested by Paulino et al. [63], based on how tourists perceive the destination.

Conversely, recent scientific research in rural tourism has concentrated on studying post-pandemic tourist activities [28,64–66], adopting a more deliberate and meaningful approach. Rural tourism has the ability to satisfy the demand of tourists seeking an escape from stress and rejuvenation in a natural environment [67], or engagement in activities focused on physical and mental well-being [66,68,69].

Areas with low tourism potential and intense agricultural utilization, where there is a high risk of negative economic impact, could have this impact reversed or at least partially halted through sustainable tourism [70].

Furthermore, there is social pressure on farmers to carry out environmentally friendly practices [71]. In such circumstances, rural tourism could serve as a viable option for farmers to adjust their businesses to preserve or even expand uncultivated zones while meeting public expectations. Certain policy proposals rooted in ecological intensification actually align with actions beneficial to rural tourism entrepreneurs [72] and the Ministry of Transport of the Slovak Republic [73]. Others have suggested tourism as a mediator between agricultural production (farming) and biodiversity conservation. Essentially, this would entail ensuring sustainable stewardship of agricultural landscapes and enhancing their versatility. This means the ability to bring numerous advantages to society beyond the production of goods [74]. Understanding the connection between land use, biodiversity, and the psychosocial factors influencing farmers to preserve semi-natural environments is crucial [75]. The role of rural tourism would be to ensure compatibility between both parties. Rural tourism, in a broader sense, allows people to enjoy the landscape and participate in agricultural activities and cultural events [71,75]. It would be a form of productive diversification, leveraging rural capital otherwise undervalued [76]. According to some scholars, agricultural tourism is perceived as a part of rural tourism [77]. Other authors, however, differentiate these concepts and their definitions based on the type of setting, the authenticity of the agricultural facility or experience, and the types of activities involved [78,79]. Nevertheless, rural tourism is generally seen as favoring the protection of the natural environment and its habitats [80].

## 2. Study Area

The region under investigation lies in the south-eastern section of the Košice region (NUTS 3), in the territory bound by the Veľké Trakany and Malé Trakany municipalities, including the alluvium in the tripoint with Ukraine and Hungary (Figure 3), as described in the works of Molokáč et al. [81] and Molokáč et al. [82]. The Zemplín Geopark area consists of an anthropogenic landscape with small riparian forests predominantly used for agriculture and housing settlements concentrated in municipalities, and, as per the portal One Soil [83], belongs to areas characterized by an average size of monoculture fields ranging from 10 to 15 ha. The geopark partially overlaps with the Ramsar Tisza Alluvium Protected Landscape Area (PLA) and encompasses features that could potentially be utilized for recreational activities and serve tourism or leisure purposes (lakes, wetlands, bogs, boundary stones, monuments). The cadastral territory of the municipalities, according to the ZB-GIS application [84], covers 3379 ha.

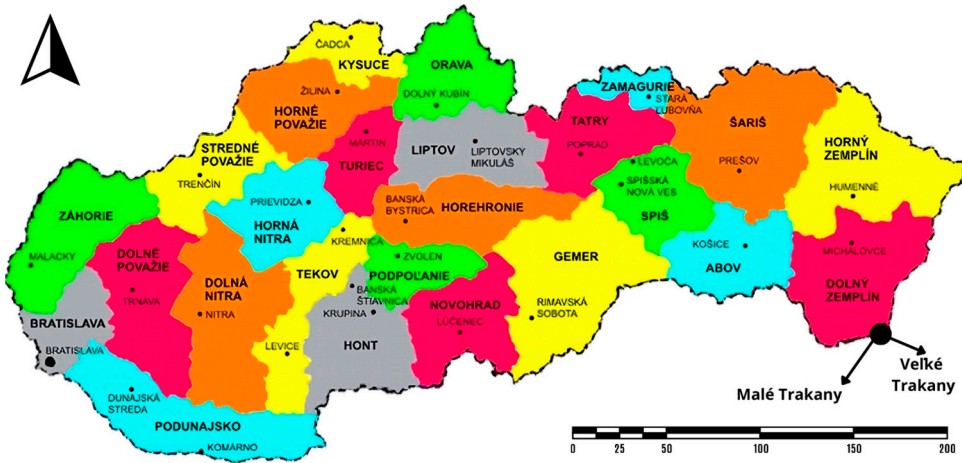

**Figure 3.** Map of Slovakia color-coded by districts (NUTS4) with the marked territory of interest. The colours reflect regional cultural similarities/differences in rural areas. Source: [85].

From a climatic perspective, the eastern part of Slovakia lies at the boundary between the maritime and continental climates of Europe, with a prevailing continental influence (subcontinental climate). In this area, northern and northeastern cold and less humid winds prevail. The natural vegetation of the territory is formed by ash, elm, oak, and alder alluvial forests, with islands of oak and hornbeam forests. In the eastern part of the cadastre near the river Tisza, however, willow and poplar alluvial forests dominate. There is a very poor presence of trees and an occasional presence of shrubs in the country. Wheat, barley, sugar beet, corn, tobacco, vegetables, lucerne, and hemp are grown there. Cherries, sour cherries, and peaches are planted in the local orchards.

## 3. Materials and Methods

This study was focused exclusively on the peripheral areas of the Slovak countryside, utilizing physical terrain reconnaissance, topographic map materials, and satellite imagery for documenting the territory under investigation. During our study, we subdivided the cultural landscape monoculture blocks into smaller units, which required measurement verification, recording, labeling, and rechecking of their size post-fragmentation. The newly formed monoculture blocks had irregular shapes (known as polygons). It was necessary to determine their dimensions, so we opted for the geodetic application GSAA for land measurement [86]. Other monitored parameters included the following: the number of fields, cultivated crops, slope inclination, overhead power lines, and the presence of landscape features (trees, tree allies, buffer strips, uncultivated green belts, perennial grass cover). In some cases, we needed to determine the length of a particular side of the land parcel, as it was intended to become part of a proposed hiking trail. For this purpose, we utilized another geodetic application, ZB GIS [84]. The obtained results were recorded in tables, which had to be checked after the overall measurement to ensure compliance with EU regulations on Good Agricultural Practices (GAP) [87], ensuring agricultural land did not exceed 30 ha. To avoid rash interventions in the landscape through the fragmentation process, we studied old map data from the 19th and 20th centuries [88,89]. In the study of historical map data, retrospective analysis methods [90] and visual interpretation methods [91] were employed. We identified old roads, which we subsequently located in the current Google Earth application [92], or at least their fragments. These roads, classified accordingly, respect the landscape in terms of geology, geomorphology, historical, and cultural aspects. We dare to assume this based on selected map data and documents from the above-mentioned period, which support claims that map data were created to serve the needs of the military, supply, and postal services, and the movement of the bourgeois population. The means of transport during that period were coaches, horses, and pedestrian traffic. Therefore, we believe that the proposed research will add

value for tourists seeking recreational and cultural activities. Some sides (borders) of the newly formed land parcels will constitute new hiking trails. The lengths of the trails were measured using geodetic tools from ZB GIS. In the final phase of the research, we had to identify sections for each hiking trail that were created by our proposal as new sections, as well as those parts that are currently used by local residents and are actively utilized. This was performed because we decided to equip these sections with green belts–windbreaks, serving as corridors for wildlife animals. Figure 4 illustrates the research diagram of our study. As depicted in the diagram, fragmentation occurred in two stages/steps of division and then continued in the third stage by presenting the results up to the present day.

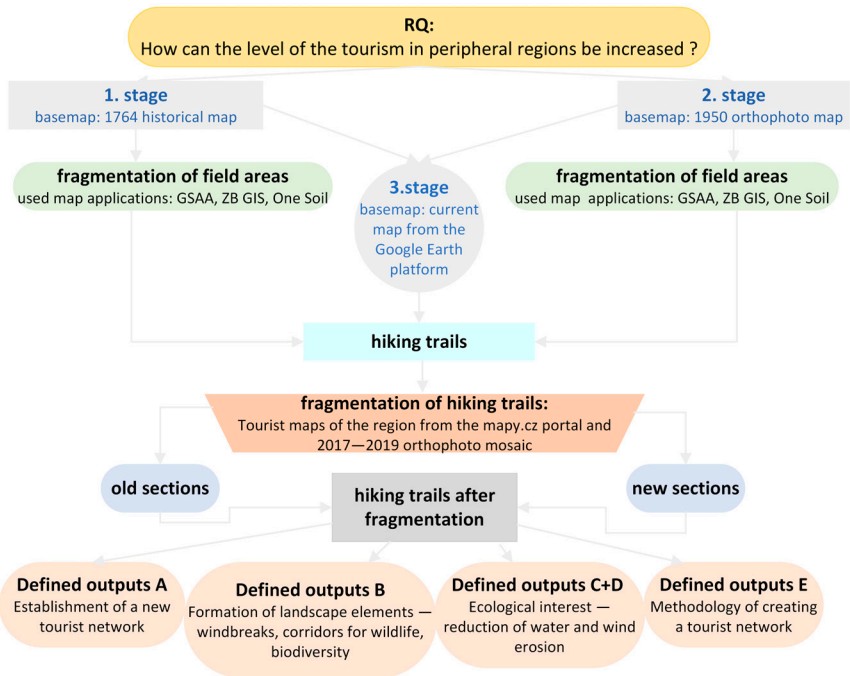

**Figure 4.** Research methodology block diagram.

### 3.1. First Stage of Landscape Fragmentation Process

The initial phase relied on data from map source No. 1 (1764–First Military Mapping), with its origin and context detailed in Jakubík's study [93]. Roads, or, to be more precise, roads of different functions and types (function—roads for carriages, army roads, and footpaths, types—narrow–sunken roads, embankment roads, bridges (made of wood and stone), along with other elements found in the landscape, have been identified.

The historical map originates from the period of the First Military Mapping in the territory of Slovakia, also known as Josephine Mapping (1764–1787). It is referred to as Josephine due to the fact that Joseph II and his mother Maria Theresa ordered a comprehensive mapping of the entire empire in 1763. This mapping was conducted "á la vue", meaning it was performed by eye estimation [88]. It involved the compilation of partial maps of individual parts of the monarchy. These maps were primarily intended for military purposes, focusing on military objects such as roads, water surfaces, bridges, fords, swamps, and springs. The depiction of the area of interest on this map is evident in Figure 5.

All roads that were identified were incorporated into the developing final map based on the Google Earth platform [92]. The portion of roads extending beyond current road infrastructure (across fields) is represented by proposed field division lines. The areas of polygons of the divided sections of the primary fields, along with the extent of the dividing sections, were determined by employing the ZB-GIS application [84], and are recorded in Table 1.

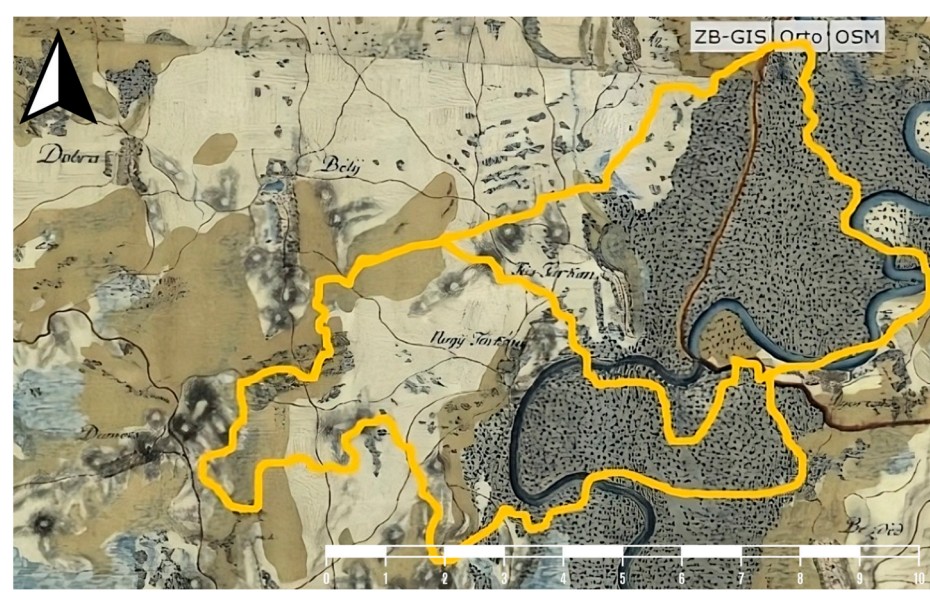

**Figure 5.** Demarcation of the area of interest on the Historical Map 1764–First Military Mapping—map source No. 1 [88].

**Table 1.** List of agricultural land acquired from the GSAA application [86] (modified by the author).

| Serial Number | Abbreviation Field Code | Field Area [ha] | Field Code |
|---|---|---|---|
| 1 | 2601/1 | 65.55 | 202127601/1 |
| 2 | 1503/3 | 64.48 | 201127503/3 |
| 3 | 2603/1 | 11.36 | 202127603/1 |
| 4 | 1603/1 | 7.73 | 201127603/1 |
| 5 | 2701/1 | 18.63 | 202127701/1 |
| 6 | 1705/1 | 3.99 | 201127705/1 |
| 7 | 1703/1 | 0.92 | 201127703/1 |
| 8 | 2705/1 | 0.41 | 202127705/1 |
| 9 | 1702/1 | 8.36 | 201127702/1 |
| 10 | 1701/2 | 22.16 | 201127701/2 |
| 11 | 1706/1 | 2.06 | 201127706/1 |
| 12 | 1707/1 | 6.96 | 201127707/1 |
| 13 | 0701/1 | 114.82 | 200127701/1 |
| 14 | 1601/1 | 28.91 | 201127601/1 |
| 15 | 0601/2 | 39.22 | 200127601/2 |
| 16 | 0507/1 | 20.21 | 200127507/1 |
| 17 | 0602/1 | 43.37 | 200127602/1 |
| 18 | 9501/1 | 147.77 | 199127501/1 |
| 19 | 9701/1 | 50.73 | 199127701/1 |
| 20 | 2702/1 | 25.03 | 202127702/1 |
| 21 | 2704/1 | 1.06 | 202127704/1 |
| 22 | 1804/1 | 0.97 | 201127804/1 |
| 23 | 2703/1 | 9.97 | 202127703/1 |
| 24 | 2805/1 | 1.28 | 202127805/1 |

**Table 1.** *Cont.*

| Serial Number | Abbreviation Field Code | Field Area [ha] | Field Code |
|---|---|---|---|
| 25 | 2801/1 | 1.03 | 202127801/1 |
| 26 | 2802/1 | 9.31 | 202127802/1 |
| 27 | 1803/2 | 18.98 | 201127803/2 |
| 28 | 2806/1 | 0.51 | 202127806/1 |
| 29 | 1802/3 | 9.19 | 201127802/3 |
| 30 | 2803/1 | 3.39 | 202127803/1 |
| 31 | 1901/1 | 141.66 | 201127901/1 |
| 32 | 0906/1 | 21.15 | 200127906/1 |
| 33 | 0901/1 | 36.86 | 200127901/1 |
| 34 | 2003/1 | 1.18 | 202128003/1 |
| 35 | 1001/1 | 0.36 | 201128001/1 |
| 36 | 2002/1 | 0.73 | 202128002/1 |
| 37 | 1902/1 | 2.45 | 201127902/1 |
| 38 | 2001/1 | 0.42 | 202128001/1 |
| 39 | 2903/1 | 1.31 | 202127903/1 |
| 40 | 2904/1 | 0.53 | 202127904/1 |
| 41 | 2902/1 | 23.61 | 202127902/1 |
| 42 | 1801/1 | 0.49 | 201127801/1 |
| 43 | 1802/3 | 9.19 | 201127802/3 |
| 44 | 0802/1 | 36.54 | 200127802/1 |
| 45 | 0804/1 | 0.68 | 200127804/1 |
| 46 | 0903/1 | 18.19 | 200127903/1 |
| 47 | 0902/1 | 1.27 | 200127902/1 |
| 48 | 0803/1 | 6.57 | 200127803/1 |
| 49 | 0801/1 | 39.4 | 200127801/1 |
| 50 | 0702/1 | 11.72 | 200127702/1 |
| 51 | 9701/1 | 50.73 | 199127701/1 |
| 52 | 9701/2 | 10.33 | 199127701/2 |
| 53 | 9802/1 | 30.33 | 199127802/1 |
| 54 | 9804/2 | 13.86 | 199127804/2 |
| 55 | 9803/1 | 1.43 | 199127803/1 |
| 56 | 9801/1 | 2.65 | 199127801/1 |
| 57 | 3701/1 | 61.81 | 203127701/1 |
| 58 | 3702/1 | 57.99 | 203127702/1 |
| 59 | 3802/1 | 15.26 | 203127802/1 |
| 60 | 3803/1 | 1.34 | 203127803/1 |
| 61 | 4704/3 | 22.54 | 204127704/3 |
| 62 | 4707/1 | 1.67 | 204127707/1 |
| 63 | 4801/3 | 96.12 | 204127801/3 |
| 64 | 5806/1 | 66.13 | 205127806/1 |

**Table 1.** *Cont.*

| Serial Number | Abbreviation Field Code | Field Area [ha] | Field Code |
|---|---|---|---|
| 65 | 5902/1 | 4.93 | 205127902/1 |
| 66 | 5903/1 | 50.86 | 205127903/1 |
| 67 | 4901/1 | 10.82 | 204127901/1 |
| 68 | 2901/1 | 5.49 | 202127901/1 |
| 69 | 3902/1 | 61.62 | 203127902/1 |
| 70 | 3901/2 | 58.25 | 203127901/2 |
| Total Field Area | | 1716.83 | |
| Average Field Area | | 24.53 | |

### 3.2. Second Stage of Landscape Fragmentation Process

The second step was based on map source No. 2 (1950 orthophoto map of Slovakia, Figure 6b). Roads, or, more precisely, the sections of roads of different types, have been identified. Some of the recognized roads matched those identified in the initial phase (intersection of the set of stage I and stage II roads). Another part of the identified roads represented a set of new/additional dividing lines (Figure 6a).

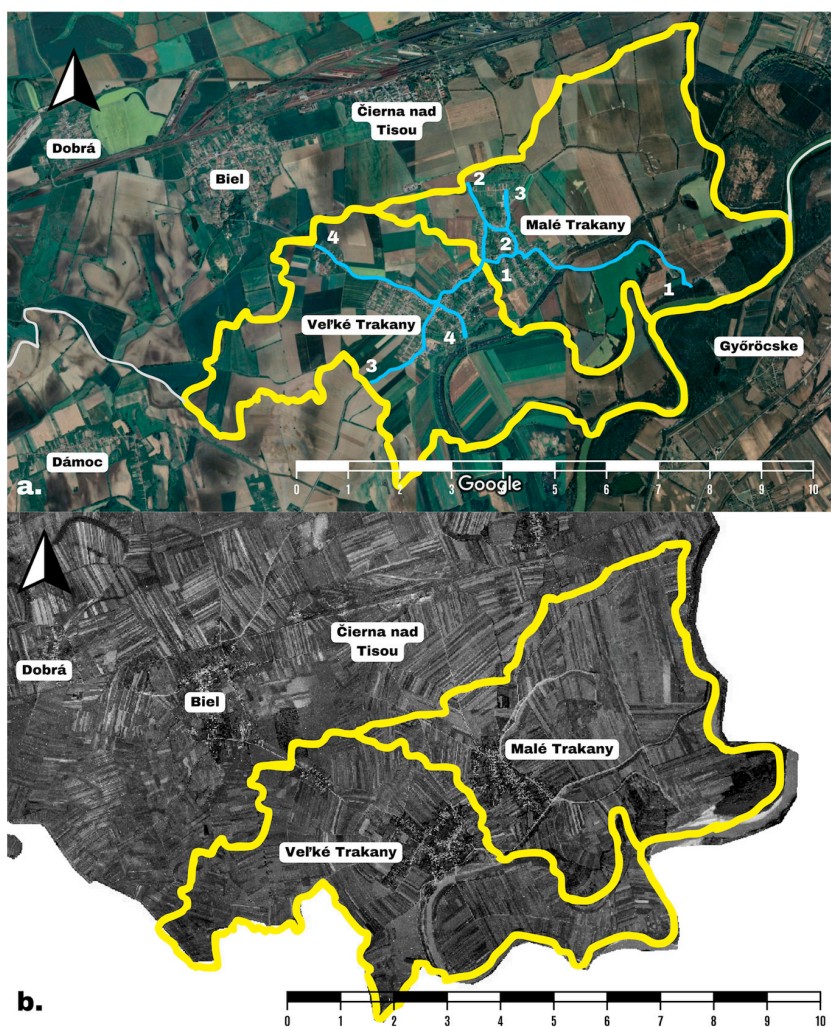

**Figure 6.** (**a**,**b**) Demarcation of the area of interest on the 1950 orthophoto map with marked municipalities (Malé Trakany, Veľké Trakany). Old (historical) roads from the military mapping are marked in yellow,

and roads identified from the 1950 orthophoto map (marked in blue) were incorporated into the current map. Each road's beginning and end were marked with numbers 1–4 in case of intersections. The boarder between Slovakia and Hungary is marked in white [89,92].

The 1950 orthophoto map of Slovakia depicts the country's territory using orthophotographic imaging, a type of map that combines topographic elevation data and photographs to present a precise representation of the Earth's surface. This specific map is created from black and white aerial photographs from the archive of the Topographic Institute in Banská Bystrica. It can be utilized for various purposes, such as monitoring the landscape due to environmental factors. The representation of the study area on this map can be seen in Figure 6.

### 3.3. Third Stage of Landscape Fragmentation Process

In the third step, work was conducted using the Google Earth mapping platform—the third map (Figure 7) reflecting the area of interest is the current map from 2023 in the Google Maps application. It is clearly visible that nearly 90% of the land area in the cadastral territories of the municipalities Veľké Trakany and Malé Trakany represents arable land (marked in purple on Figure 7).

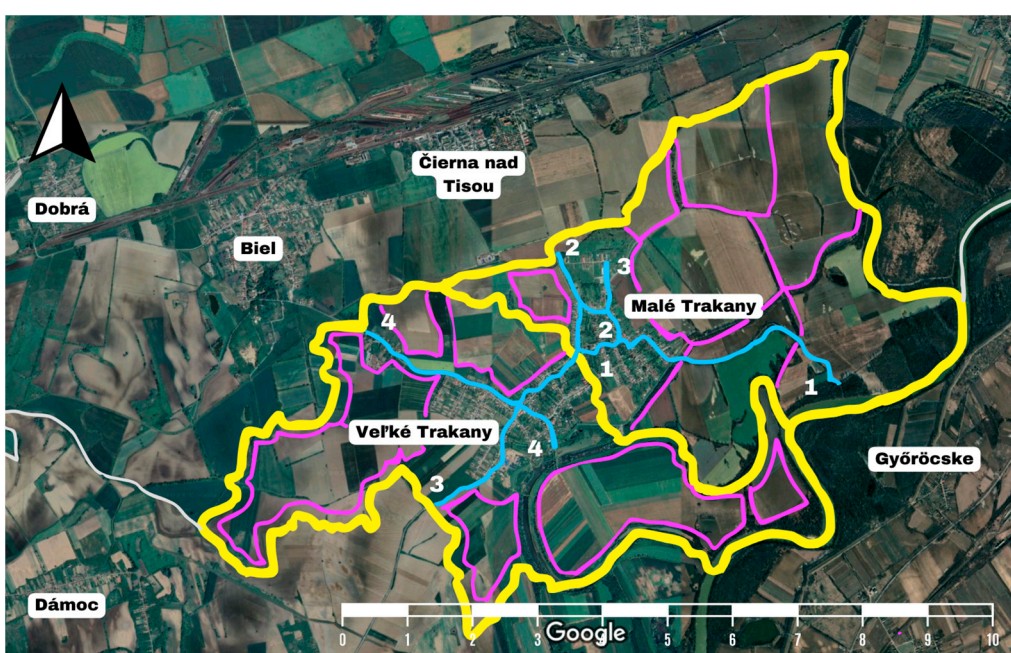

**Figure 7.** Demarcation of the area of interest in the Google Earth mapping platform, with arable land marked in purple. Each road's beginning and end were marked with numbers 1–4 in case of intersections. Source: processed by the author using Google Earth [92].

The result of the described steps includes three map layers capturing fragmentation at each stage. Following this, parameters were assessed for every stage, including the following:

- Average field sizes, dimensions of individual fields post-division, quantity of fields, quantity of divided/formed fields, and length of dividing (proposed) lines;
- Level of similarity of roads from various historical eras and alternatives for designed dividing lines, i.e., tree-lined alleys (various kinds of woody plants) and buffer strips.

The proposal was developed following the technical guidelines of road-side vegetation management as defined by the Slovak Ministry of Transport, Post and Telecommunications [94].

Based on the various implementation proposals, the outputs were determined as follows:

(a)  Length of accessible trails for so-called 'soft' form of tourism in the sense of Bacsi & Tóth [60] and Weaver & Lawton [95];
(b)  Organic matter (biomass) production;
(c)  Reduction of fields' sections susceptible to water erosion;
(d)  Reduction of fields' sections prone to wind erosion;
(e)  Quantity of honey bee colonies that can be maintained.

## 4. Results and Discussion

Referring to the assessment of mapping data [86] and field research, a detailed delineation of the study area was carried out by defining its geopolitical boundaries (border with Hungary, border with Ukraine) and delineation of borders along field or forest roads to ensure that the study area's borders did not intersect intact fields (Figure 8). The study area spanned 3379 hectares and comprised a total of 98 fields. The boundaries of the study area are highlighted in yellow in the figure.

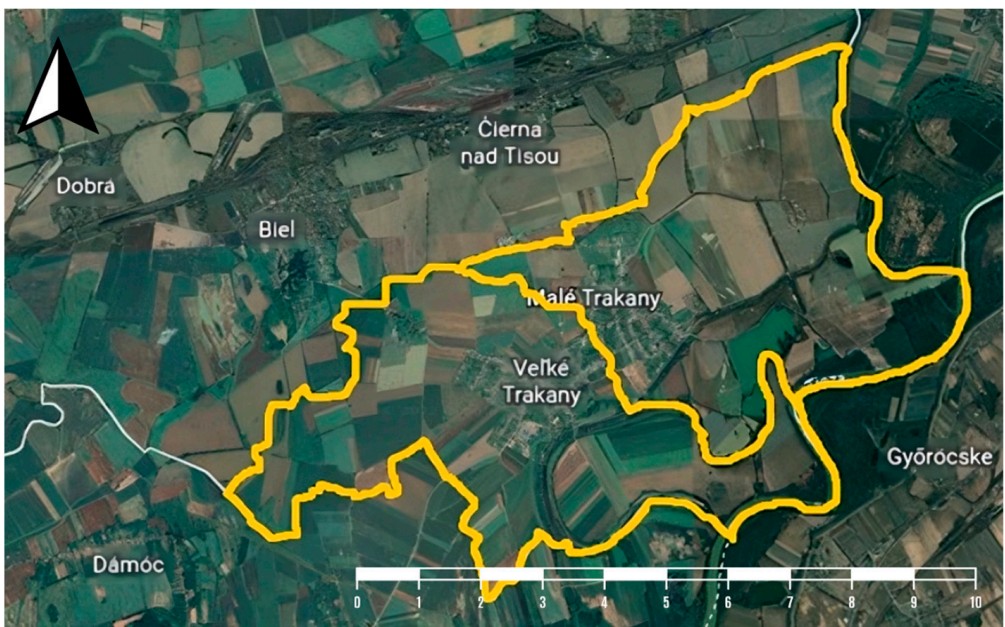

**Figure 8.** Demarcation of the study territory. Boundary lines are formed by natural features (Malé Trakany, Veľké Trakany). Source: processed by the author using the online Google Earth map [92].

If we exclude fields located within the built-up area (gardens), the total area under analysis is 2980.34 ha. Within this area, there are 70 fields, with 20 having an area greater than 30 ha. The size of the largest and smallest fields ranges from 147.77 to 0.36 hectares. The average field size of all fields is 24.53 hectares.

The documentation of fields was conducted through identification according to the LPIS system. This approach allows for the specific identification of each field on the map as needed, as well as the assignment of additional attributes to each field.

*First and Second Stages of Landscape Fragmentation Process*

In the first step, based on the data from the 1764–1787 Military (Mapping) Survey, 10 roads with a total length of 12.37 km were discerned (Figure 7). In the second step, based on the map data from the 1950 orthophoto map, four roads covering a total of 12.37 km were recognized (Figure 9).

In the initial phase, four fields were divided, covering an area of 365.72 ha (Table 2), and in the following phase, 15 fields were divided, covering an area of 897.79 ha (Table 3). Altogether, 19 and 46 new fields were formed.

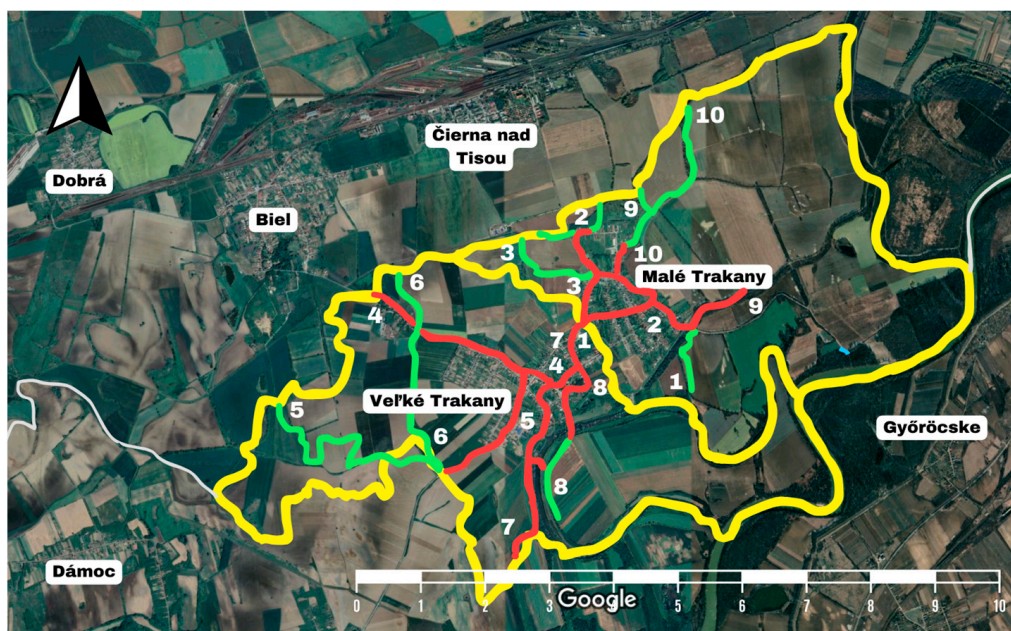

**Figure 9.** Dividing lines/roads of both stages of division. The marking is as follows: first stage—green, second stage—red. In case roads intersected, the beginning and end of each road were additionally marked with numbers 1–10. Source: processed by the author using the online Google Earth map [92].

**Table 2.** List of divided agricultural land parcels according to old roads from the first mapping [88].

| Serial Number | Original Field Area [ha] | Field Designation | Number of Resulting Fields | Resulting Field Designation (Designation of Newly Created Parcels after Division) | Resulting Field Area [ha] | Length of Dividing Line [km] |
|---|---|---|---|---|---|---|
| 1 | 61.81 | 3701/1 | 4 | A | 7.20 | 0.47 |
| | | | | B | 25.71 | 0.55 |
| | | | | C | 13.90 | 0.46 |
| | | | | D | 15.00 | |
| 2 | 141.66 | 1901/1 | 7 | A | 26.50 | 1.14 |
| | | | | B | 14.00 | 1.32 |
| | | | | C | 17.70 | 0.92 |
| | | | | D | 15.00 | 0.42 |
| | | | | E | 17.00 | 0.62 |
| | | | | F | 24,60 | 0.62 |
| | | | | G | 26.70 | |
| 3 | 96.12 | 4801/3 | 5 | A | 23.50 | 0.60 |
| | | | | B | 20.40 | 0.81 |
| | | | | C | 18.00 | 0.84 |
| | | | | D | 15.40 | 0.40 |
| | | | | E | 18.70 | |
| 4 | 66.13 | 5806/1 | 3 | A | 25.90 | 0.70 |
| | | | | B | 19.20 | 0.45 |
| | | | | C | 20.80 | |
| Total Field Area | 365.72 | | 19 | | 365.21 | 10.30 |
| Average Field Area | 91.43 | | | | 91.3 | |

**Table 3.** List of divided agricultural land parcels according to old roads from the second mapping [89].

| Serial Number | Original Field Area [ha] | Field Designation | Number of Resulting Fields | Resulting Field Designation (Designation of Newly Created Parcels after Division) | Resulting Field Area [ha] | Length of Dividing Line [km] |
|---|---|---|---|---|---|---|
| 1 | 65.55 | 2601/1 | 3 | A | 23.60 | 0.62 |
| | | | | B | 26.50 | 0.64 |
| | | | | C | 15.50 | |
| 2 | 114.82 | 0701/1 | 5 | A | 28.40 | 1.08 |
| | | | | B | 27.10 | 1.04 |
| | | | | C | 16.00 | 1.00 |
| | | | | D | 20.50 | 0.96 |
| | | | | E | 22.90 | |
| 3 | 39.22 | 0601/2 | 2 | A | 20.60 | 0.53 |
| | | | | B | 18.60 | |
| 4 | 43.37 | 0602/1 | 2 | A | 17.90 | 0.49 |
| | | | | B | 25.50 | |
| 5 | 64.48 | 1503/3 | 3 | A | 17.20 | 1.00 |
| | | | | B | 25.50 | 0.46 |
| | | | | C | 21.78 | |
| 6 | 147.77 | 9501/1 | 8 | A | 12.50 | 0.53 |
| | | | | B | 21.90 | 0.60 |
| | | | | C | 24.50 | 0.59 |
| | | | | D | 21.00 | 1.15 |
| | | | | E | 19.50 | 0.31 |
| | | | | F | 9.67 | 0.93 |
| | | | | G | 24.20 | 0.32 |
| | | | | H | 14.50 | |
| 7 | 50.73 | 9701/1 | 3 | A | 4.93 | 0.27 |
| | | | | B | 21.00 | 0.43 |
| | | | | C | 24.80 | |
| 8 | 36.86 | 0901/1 | 3 | A | 18.26 | 0.58 |
| | | | | B | 18.60 | |
| 9 | 36.54 | 0802/1 | 2 | A | 14.80 | 0.51 |
| | | | | B | 21.70 | |
| 10 | 39.40 | 0801/1 | 2 | A | 20.80 | 0.46 |
| | | | | B | 18.60 | |
| 11 | 30.33 | 9802/1 | 2 | A | 18.80 | 0.40 |
| | | | | B | 11.53 | |
| 12 | 57.99 | 3702/1 | 3 | A | 20.40 | 0.82 |
| | | | | B | 22.80 | 0.86 |
| | | | | C | 14.30 | |
| 13 | 50.86 | 5903/1 | 2 | A | 23.56 | 0.66 |
| | | | | B | 27.30 | |
| 14 | 61.62 | 3902/1 | 3 | A | 26.90 | 1.38 |
| | | | | B | 19.70 | 0.48 |
| | | | | C | 14.80 | |
| 15 | 58.25 | 3901/2 | 3 | A | 25.00 | 1.11 |
| | | | | B | 9.62 | 0.46 |
| | | | | C | 23.50 | |
| **Total Field Area** | 897.79 | | 46 | | 897.05 | 20.67 |
| **Average Field Area** | 59.85 | | | | 59.8 | |

Figure 10 illustrates the graphical representation of the average field size at the initial phase and subsequent fragmentation phases.

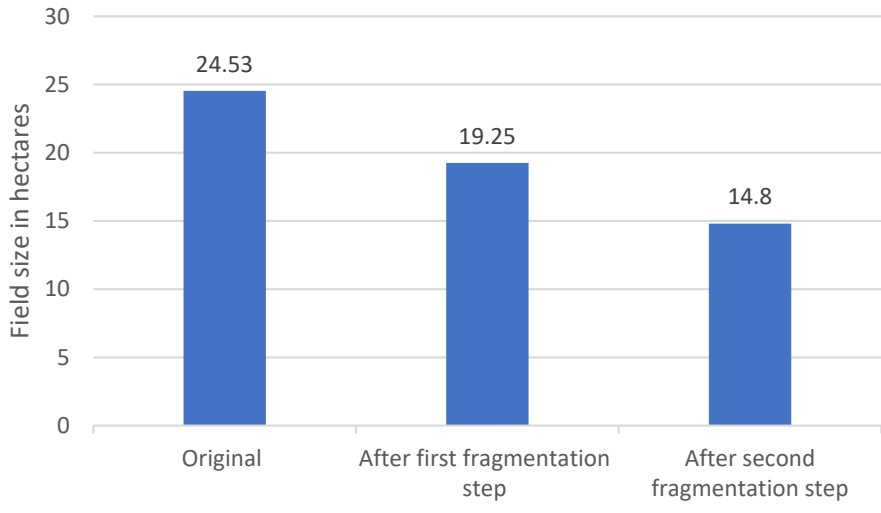

**Figure 10.** The comparison of the initial average size of the field to the average field sizes after the first and second fragmentation phases.

The quantity of fields with sizes exceeding 30 hectares was 15 after the first phase and 0 after the second phase. The percentage of divided fields at each division level can be seen in Figure 11.

PERCENTAGE OF DIVIDED FIELDS FOR INDIVIDUAL
LEVELS OF DIVISION [%]

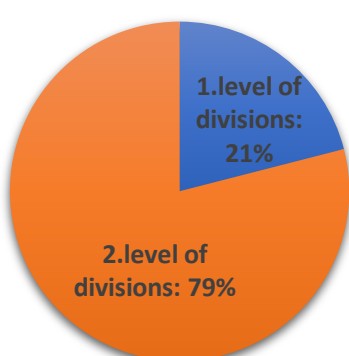

**Figure 11.** Percentage of divided fields for individual levels of division.

In certain sections of the roads identified during the individual division steps the roads aligned with field borders, thereby eliminating the need for additional dividing lines. Hence, the combined lengths of the dividing lines are smaller than those of the individual roads transferred from the original map sources.

After the first stage of division, the combined length of the dividing lines was 10.30 km, and it was 20.67 km after the second stage of division. The number of dividing lines corresponding to individual steps is depicted in Figure 12 and Tables 4 and 5.

### NUMBER OF DIVIDING LINES CORRESPONDING TO INDIVIDUAL DIVISION STEPS

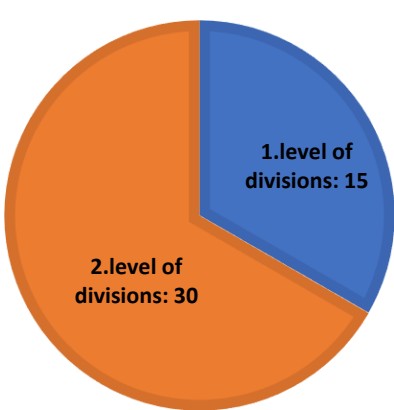

**Figure 12.** Number of dividing lines corresponding to individual division steps.

**Table 4.** Historical/old roads from the years 1764–1787.

| Serial Number | Direction | Length of the Road | Length of the Section through the Field |
|---|---|---|---|
| 1 | From Tisza River to Malé Trakany | 3.063 km | 2.145 km |
| 2 | From Malé Trakany to Čierna nad Tisou | 3.477 km | 1.124 km |
| 3 | From Čierna nad Tisou to Malé Trakany | 2.322 km | 1.167 km |
| 4 | From Veľké Trakany to Biel | 3.495 km | 1.829 km |
| 5 | From Veľké Trakany to Hungary | 4.332 km | 3.551 km |
| 6 | From Biel to Hungary | 2.362 km | 2.362 km |
| 7 | From Veľké Trakany to Hungary | 3.403 km | 3.403 km |
| 8 | From Veľké Trakany to Hungary | 4.123 km | 4.123 km |
| 9 | From border area of Čierna nad Tisou to Tisza River | 2.861 km | 2.861 km |
| 10 | Malé Trakany | 1.549 km | 1.549 km |

**Table 5.** Historical/old roads from the year 1950.

| Serial Number | Direction | Length of the Road | Length of the Section through the Field |
|---|---|---|---|
| 1 | From Malé Trakany to Tisza River | 3.763 km | 0 km |
| 2 | From Malé Trakany to Čierna nad Tisou | 3.008 km | 0 km |
| 3 | From Malé Trakany to Hungary | 3.119 km | 0 km |
| 4 | From Veľké Trakany to Biel | 2.480 km | 0 km |

Defined outputs:

(a)

In the context of the present era, as efforts are made to mitigate over-tourism and to transition to "soft" alternatives, a significant aspect of the proposal involves establishing a network of tourist routes. We categorize the proposed trails as "soft" due to their easy difficulty level. These routes aim to enhance accessibility to notable landmarks in the landscape (e.g., biocenter with floodplain forests near the Tisza, boundary stones, monuments). Additionally, they can serve as areas for recreational activities and countryside stays. In total, 14 trails were proposed in the study. The combined length of the tourist routes that emerged during the fragmentation process in the first and second stages is 35.44 km, interconnecting two municipalities.

Based on the obtained results in the form of new hiking trails and their designated lengths, consideration was given to their use to kickstart soft tourism in an area where any form of tourism is entirely absent. According to Jiří Vaníček [96], the calculation of the attractiveness of tourist destinations (soft hiking trails) depends on three premises, and primarily on the number of visitors (residents). Statistical monitoring of social and economic indicators is not conducted at the LAU 2 level in the Slovak Republic. In further research, we propose implementing several methodological approaches, the combination of which will ensure the acquisition of a comprehensive set of data. Comparable research was undertaken in the Czech Republic during the construction of single trails in a forest environment [97]. Visitor numbers could be determined through systematically placed automatic counters in each area to cover the overall resident visitation within a given time interval or the use of location telecommunication data. The next steps would involve qualitative (semi-structured interviews and ethnography) and quantitative (survey questionnaire technique) data collection methodologies. We can rely on data from the Statistical Office of the Slovak Republic for local residents, of whom there are 2619 directly in the selected area [98]. The average age of residents is 39.6 years (data as of 1 January 2021) [99]. Alternatively, we suggest monitoring visitor numbers using the methodology of quantifying tourism intensity [100] using the tourism intensity index (TII). Other methodological issues of detecting tourist attendance data are being discussed, e.g., Mariot [101], Andersen et al. [102], Šveda et al. [103].

In light of the findings, i.e., the lengths of the dividing lines, hiking trail construction and subsequent maintenance were considered as, before the proposal, the territory of interest was not a tourist area. The marking methodology would comply with thle vaid Slovak guidelines for marking hiking trails and placing noticeboards that are included in the STN 01 8025 standard [104]. The financing of soft hiking trail management and maintenance should have an operator, which could be one of the concerned municipalities. The operator, as a tourism entity, cannot completely satisfy visitors' needs in the destination without the help of the other two actors (facility operators and visitors), Gummesson [105]. Potentially, the operator can also be a local partnership known as a local action group (according to the Leader approach, Rural Development Policy of the European Union) [106]. The funding of soft hiking trails will require multiple sources, consisting of government subsidies (subsidies of the Košice Region), corporate partnership sources, other public sources (the budget of the city of Trebišov), and a financial contribution from the Ministry of Transport and Construction of the Slovak Republic that is to be applied for. Volunteering activities organized by local communities will be essential for soft hiking trail maintenance. The trail maintenance-focused activities attended by both residents and visitors from the surrounding area should be held at regular (monthly) intervals. This way, the operator can reduce its operating costs.

(b)

When planting vegetation, i.e., bushes and trees, the weather conditions and the variety of species present in the given location should be considered, and, in this case, hawthorn bushes and primarily trees such as winter oak, wild pear, and wild chestnut [107] would be the right choice. Single or multiple rows of trees and trusses, with branches extending from the ground level to the tops of the trees, could be used as shelterbelts. When using vegetation as windbreaks, it is necessary to ensure that it is perpendicular to the prevailing wind direction.

The total number of woody plants taller than 3 m and spaced 3 m apart amounts to 30,970 units. For shrubs with a height of up to 1.5 m and a spacing of 1.5 m, the count is 61,940 units, and, with a spacing of 1 m, it is 92,910 units. In the case of a single-row planting of trees with a spacing from 8 m to 6 m, the count ranges from 3871 to 5162 units. For a double-row, such as a two-sided alley along the road with the same spacing, the count ranges from 7742 to 10,324 units. For a triple-row, which can be accommodated in a 22-meter-wide strip, the counts range from 11,613 to 15,486 units. According to the Slovak Ministry of Transport, Post and Telecommunications [108], the establishment of vegetation in a 22-meter-wide green belt (vegetative barrier) is capable of acting as a safeguard for agricultural land against aeolian erosion and soil degradation, contingent upon environmental factors [109,110].

(c) + (d)

The historical roads were intentionally led through the country, typically along linear, frequently elevated, well-drained areas away from muddy sections, etc. According to Bolina et al. [111], this fact contributes significantly to the protection against soil erosion. When designing the roads, the objective was to minimize physical activity when overcoming height differences (both up and down) by people and draft animals.

Boundary lines proposed for the western part of the territory under consideration were designed to eliminate the north and southeast wind flow by dividing the sections in a northeast-to-southwest direction. When implementing and refining the design of buffer strips, it is essential to consider the requirements of current agricultural technology, particularly regarding the questionable suitability of widely adopted technology concerning its operation and transportation, as well as the connectivity and accessibility of all the fields. From the viewpoint of farmers, the additional fuel costs and the necessity for extended working hours (number of passes and downtimes) may occur in such a fragmented landscape. These aspects need thorough consideration, analysis, and incorporation into forthcoming laws or regulations.

Considering the proper integration of vegetation, like trees and bushes, the proposed arrangement of the dividing lines aims to prevent the plowing of the ecologically significant area, thereby preserving its natural vegetation. We agree with the findings of the Polish scientists P. Postek, P. Leń, and Ż. Stręk [112], which state that, although there are road network databases, there are no data on the transitions between urban and agricultural land available (Topographic Objects Database—BDOT database), and there is a lack of data defining the minimum width of roads that would ensure the trouble-free movement of agricultural machinery.

(e)

From the perspective of ensuring enough food for pollinators, especially bees, it is possible to count on one bee colony per 10 ha, representing a total of seven bee colonies, according to the local conditions [113].

Our contribution lies in the provision of the absent methodology that would form the basis for agricultural land fragmentation. During this process, one must consider the shortcomings of the legal regulations related to inheritance law and other estate-related legal issues concerning land ownership. This is a throwback to past times, which is very clearly described by Sklenička et al. [114]. The above results in the improper functioning of tourism in Slovakia and in unsettled estate ownership relations which, in turn, prevent investments in rural infrastructure, environmental facilities, and measures necessary for protecting natural resources, including biodiversity and adaptation to climate change [115].

The Ministry of Agriculture and Rural Development of the Slovak Republic is introducing projects changing the layout of parcels so as to settle ownership relations, as the unsettled plots are the main obstacle to rural development.

Furthermore, our methodology proposes the multifunctional use of the agricultural countryside, which no longer fulfills its primary function. This way, the social pressure put on farmers would bring a new direction to the use of the parts of farmed land that are not suitable for agricultural purposes. Thus, their use will correspond with the solution for forming the Intervention Strategy of the 2023–2027 CAP Strategic Plan [115]. Through the LEADER program [106], which is an intervention strategy for young farmers, and NATURA 2000, both forming part of the Strategy, viable agriculture has been encouraged even in the disadvantaged areas of the Slovak Republic. The above will then lead to an increased level of rural tourism activities. The same applies to other Slovak regions, as can be seen, for example, in the work by Gregorová & Hrončak [116], which focuses on Central Slovakia. Beresecká & Varecha [117] have studied the causes of the unfavorable position of tourism in Slovakia and the Nitra Region and have concluded that the cause lies in the underutilization of rural tourism products and of the territory's unevenly distributed potential. Gregorová & Korec [118] have also focused on rural tourism in the eastern part of Slovakia, where the possibilities and limitations of tourism development have been examined.

Practically speaking, all the works evaluating the regional structure of Slovakia based on social (socio-economic) development state that the Banská Bystrica Region (GDP per capita in PPS was 55% of the EU average in 2014), the Prešov Region (46% of the same), and the Košice Region (60% of the same) are the underdeveloped NUTS3 Slovak regions [119–128]. The most lagging region of Slovakia on the NUTS2 level is East Slovakia (53%) [129]. The mentioned authors pointed out that the Eastern Slovak Region is behind the Slovak average and the rest of the Slovak NUTS2 regions.

The country could thus benefit from the preservation of its natural character by planting native woody plants, trees, and shrubs. Should our research be implemented it could significantly support the greening of the environment (by planting woody plants), prevent hydraulic and aeolian erosion (the creation of windbreaks), and reverse the biodiversity loss in the country (bees, predators, insects, pollinators).

## 5. Conclusions

Our aim was to identify possible solutions to meet the commitments and directives set forth by the EU within the Slovak environment regarding landscape fragmentation while leveraging the multifunctionality of the former agricultural countryside. The preparation of the Strategic Plan for Rural Development 2021–2027 and the goals of the 2030 agenda focus on the matter of utilizing the cultural landscape. Within the framework of Slovakia, this means interventions in an environment that is economically and socially disadvantaged. If implemented, our research could establish the methodology according to which large-scale monoculture fields would be fragmented, as stipulated in the EU's good management practices.

From a pessimistic point of view, it is anticipated that changes may fail to yield the desired outcomes and could be considered unjustified. Conversely, an alternative approach involves seeking solutions with long-term positive effects, benefiting not only local residents directly impacted by the changes but also those who may experience their effects indirectly. The common primary goal will be to increase tourism within the local natural environment through the establishment of hiking trails, while conscientiously considering various historical and ecological factors. It is crucial for the resulting hiking trails to be functional and to enhance the aesthetic image of the countryside, as such, they should not just be thoughtlessly introduced into the landscape. This highlights the need to protect and further utilize one's surroundings in a sustainable manner. Secondary objectives will include compliance with EU regulations on biodiversity protection, reduction of carbon footprint, and erosion prevention. Key potential benefits also include the psychological and aesthetic effects, the variability of the country, and the isolation function (i.e., suitable living conditions for animals).

The contribution of our work is further seen in addressing the tourism development challenges, particularly in regions where some forms of tourism (albeit at a minimal level) are already present. We aim to enhance these areas by introducing rural tourism as a new facet. We anticipate a gradual increase in tourists in the region due to the expanded range of services, which will have a positive impact on economic indicators and will enhance local residents' satisfaction. In the future, the problem of meeting EU commitments could be further addressed by solving two identifiable issues: (a) property rights and (b) inadequate recording of map data. There is the potential to map as much of Slovakia's territory as possible to prevent insufficient and incomplete fragmentation of specifically delineated areas. It will be interesting to tackle this problem and also to analyze the solutions of other post-communist countries in the EU. Additionally, in future research endeavors, we would also like to provide a comparative analysis of cultural landscape fragmentation that encompasses countries beyond Europe.

**Author Contributions:** Conceptualization, J.R., R.R., D.T. and G.W.; methodology, J.R. and R.R.; formal analysis, D.T.; investigation, J.R.; resources and data curation, J.R., R.R., D.T. and G.W.; writing—original draft preparation, J.R., R.R., D.T. and G.W.; writing—review and editing, D.T.; visualization, J.R. and R.R.; supervision, J.R.; funding acquisition, G.W. All authors have read and agreed to the published version of the manuscript.

**Funding:** This study was funded by Grant Agency of the Ministry of Education, Science, Research and Sport of the Slovak Republic, KEGA project no. 056TUKE-4/2024, "A platform for the effective creation, assessment, and transfer of innovations and effective handling of the results of scientific research activities of universities with a focus on practice".

**Institutional Review Board Statement:** Not applicable.

**Informed Consent Statement:** Not applicable.

**Data Availability Statement:** Data are contained within the article.

**Conflicts of Interest:** The authors declare no conflicts of interest.

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
