# Peer review of "The Use of Cultural Landscape Fragmentation for Rural Tourism Development in the Zemplín Geopark, Slovakia"

_sustainability, doi:10.3390/su16104011_

Round 1

Reviewer 1 Report (Previous Reviewer 5)

Comments and Suggestions for Authors

The authors have addressed reviewer's concerns and is ready to proceed for publication.

Comments on the Quality of English Language

English is fine

Author Response

Dear reviewer,

Thank you for all your previous comments and recommendations that have helped us advance the quality of our paper.

Reviewer 2 Report (Previous Reviewer 2)

Comments and Suggestions for Authors

Manuscript number: sustainability-2993572

Title: The Use of Cultural Landscape Fragmentation for Rural Tourism Development in the Zemplín Geopark, Slovakia

General comments:

It is a resubmitted manuscript (former Manuscript-ID: sustainability- 2851884). The author team has put a lot of efforts into the first draft of the original manuscript (sustainability- 2851884), processed and improved the overall level of this second draft of the manuscript (sustainability-2993572).

The author mentions that the purpose of this article is to propose a solution for the issue of natural resource renewal and the establishment of sustainability in the studied area. Unfortunately, I don't see the approach as innovative but rather as a reinvention and description of previous approaches. The ‘Introduction’ is too long. In fact, there is no need to include so many references, and a separate literature review is unnecessary, as these minor errors greatly impact the overall quality of the MS.

Specific comments:

(1) Line 135-237, this section is too long! Please simplify them.

(2) Line 236-237, the sentence seems to contain a grammatical error.

(3) Line 244, please take note of the document code format.

(4) Line 305, the picture quality is not clear, please reduce the size of the picture. The same applies to the other graphs.

(5) Line 420-426, the width and height of the table rows are too small; please adjust them.

(6) Line 429, add axes to the graph for a more formal presentation.

(7) Line 518, it is suggested that the author should combine these two parts into one.

Author Response

Reviewer 3 Report (Previous Reviewer 4)

Comments and Suggestions for Authors

Many thanks to the authors. The paper is well improved and the authors have responded to all commentaries of the review. The paper is well structured, the methodology and research results are clearly described and properly analysed, and discussed.  The list of the references is impressive, the authors have referred to recent and international studies.

I look forward to seeing the publication.

Author Response

Dear Reviewer,

Thank you very much for your positive feedback and for all your previous comments and recommendations that have helped us advance our paper.

This manuscript is a resubmission of an earlier submission. The following is a list of the peer review reports and author responses from that submission.

Round 1

Reviewer 1 Report

Comments and Suggestions for Authors

This paper tried to use the cultural landscape fragmentation for rural tourism development. The scientific importance is high as I believed, which is a very classic topic in the field of rural geography studies. However, the whole manuscript is challenging to understand and lack of some major developments. The overall quality of the manuscript needs to make significant revisions, especially in related work review, and methodology design.

1.     While the author furnishes a comprehensive and systematic overview of rural tourism and the Slovenian context, the content predominantly revolves around theoretical knowledge. I suggest incorporating quantitative studies, particularly exploring how Cultural Landscape Fragmentation influences rural tourism. Consider exploring quantifiable indicators or methods for characterizing Landscape Fragmentation.

2.     Concerning the proposed methodological framework, the rationale behind this approach appears somewhat unclear. Despite the author providing numerous table entries, further clarification is needed regarding the logic behind method selection. I recommend the author provides additional insights into the rationale for choosing these methods.

3.     The citation style for the references in the paper is somewhat disorganized, featuring a mix of author-year and numerical methods. Please standardize the citation format throughout the manuscript to ensure uniformity.

4.     The clarity of the images is suboptimal, and it is recommended to increase the resolution to 600 dpi for better visual quality.

Author Response

Dear reviewer,

thank you for your time and for all the comments and suggestions that will help improve our manuscript. Responses to all comments and suggestion are attached in cover letter. 

Reviewer 2 Report

Comments and Suggestions for Authors

Manuscript number: Sustainability-2851884

Title: The Use of Cultural Landscape Fragmentation for Rural Tourism Development in the Zemplín Geopark, Slovakia

General comments:

The author mentions that the purpose of this article is to propose a solution for the issue of natural resource renewal and the establishment of sustainability in the studied area. Unfortunately, I don't see the approach as innovative but rather as a reinvention and description of previous approaches. Furthermore, the entire article suffers from significant formatting issues, and these minor errors greatly impact the overall quality of the article. I suggest that the author should refer to high-quality literature and carefully make changes to both format and content chart of this article. Given the influence of the Journal "Sustainabilty", I recommend that this manuscript be thoroughly revised.

Specific comments:

1) Article structure

The article lacks clarity in its structure, and it is suggested to merge the “Introduction” with the “Theoretical Background” into a new section.

2) Figures and Tables

All figures are not  clear enough, so it is recommended to modify them according to the journal’s requirements. Pay special attention to standardizing the geographical area map by including a scale, beacon, and legend. Please strictly modify these figures according to the specified requirements. In addition, the table border is inconsistent in thickness, e.g., Table 1. Please revise them.

Additional comments:

1) Conclusion: This section is too long and verbose! You have already mentioned the purpose of the study in the “method” section, so there is no need to emphasize it here. 

2) Reference: Please revise the reference in accordance with the journal’s formatting requirements.

Author Response

Dear reviewer, 

Thank you for all your suggestions. Responses to all your comments are attached in cover letter. 

Reviewer 3 Report

Comments and Suggestions for Authors

1.The introduction part provides a detailed introduction to Slovakia's existing research, but considering that the publication of the article is international, it is recommended to supplement the current status and existing problems of relevant research in other countries.

2.When introducing the theoretical background, it is recommended to draw a framework diagram that can provide readers with a clearer presentation.

3.There is an issue with the numbering of the titles at each level in the fourth part of the article, materials and methods. It is recommended to adjust it again.

4.The results and discussion in the article are quite extensive and confusing. It is recommended to separate the results and discussion. Additionally, if allowed, it is recommended to include some charts in the appendix.

5.The conclusion generally does not require reference to literature. If further analysis is needed, it is recommended to include it in the discussion.

6.The figures used in the text are too rough and have low resolution, and need to be redrawn.

Author Response

Dear Reviewer, 

Thank you for all your suggestions. Responses to all your comments could be found in attached cover letter. 

Reviewer 4 Report

Comments and Suggestions for Authors

Very well structured and written paper. As the authors state, the aim of the research is to propose a solution to the issue of natural resource renewal and the establishment of sustainability in the studied area. However, there is no clear research questions or hypothesis defined. This should be included to get the paper more focused. Also later in the data analysis part and conclusions the authors should reflect on the approval/ rejection of hypothesis or give the answers on the research questions.

The authors have used more than 120 literature sources for  their theoretical analysis. They are diverse, relevant and up-to-date.

The authors have well described the research methodology and the logic of research. Also data analysis present the main results and are well elaborated. Pictures, however, are not of very high quality. This should be improved to increase the overall quality of the paper.

Author Response

Dear Reviewer, 

Thank you for all your comments. Responses to your suggestions are attached in cover letter. 

Reviewer 5 Report

Comments and Suggestions for Authors

1. Some terminologies are not academic, such as soft hiking trails, soft tourism, etc. It might require further explanation/justification.

2. Line 119, Sec 2. Theoretical Background, not sure if the context is pertinent to the subtitle.

3. Line 177, 'rural (sustainable) tourism' confuses, for rural tourism and sustainable tourism are two different concepts. Might be related but they are not the same.

4. The study is based on a series of stages of landscape fragmentation processes, is it about area fragmentation or route fragmentation, require further explanation.

5. The study intended to bring a new way of looking at agriculture as a tourism product, which is already a substantial discipline in the tourism sector. Therefore it is suggested to the authors to incorporate more existing literature in the manuscript.

Comments on the Quality of English Language

The quality of the English language is fine.

Author Response

Dear Reviewer, 

Thank you for your suggestions. Our responses to all your comments are in attached cover letter.
